# Automatic Identification System for Rock Microseismic Signals Based on Signal Eigenvalues

Junzhi Chen [1], Hongbo Li [1], Chunfang Ren [1,*] and Fan Hu [2]

1 Faculty of Land Resources Engineering, Kunming University of Science and Technology, Kunming 650031, China
2 Changsha Institute of Mining Research Co., Ltd., Changsha 410012, China
* Correspondence: 20090038@kust.edu.cn

**Abstract:** The microseismic signals of rock fractures indicate that the rock mass in a particular area is changing slowly, and the microseismic signals of rock blasting indicate that the rock mass in a particular area is changing violently. It is of great significance to accurately distinguish rock fracture signals and rock microseismic signals for analyzing the changes in the rock mass in the area where the signal occurs. Considering the microseismic signals of the Dahongshan Iron Mine, the time domain, frequency domain, energy characteristic distribution, and fractal features of each signal were analyzed after noise reduction of the original signal. The results demonstrate that the signal duration and maximum amplitude of the signal could not accurately distinguish the two types of signals. However, the main frequency of the rock fracture signal after noise reduction is distributed above 500 HZ, and the main frequency of the rock blasting signal is mainly distributed below 500 HZ. After the denoised signal is decomposed by the ensemble empirical simulation decomposition, the energy of the IMF1 frequency band of the rock fracture signal occupies an absolute dominant position, and the sum of the energy of the IMF2–IMF4 frequency bands of the rock blasting signal occupies a dominant position. The fractal box dimension of the rock fracture signal is mainly below 1.1, and the fractal box dimension of the rock blasting signal is mainly above 1.25. According to the above research results, an automatic signal recognition system based on the BP neural network is established, and the recognition accuracy of the rock blasting and rock fracture signals reached 93% and 94% respectively, when this system was used.

**Keywords:** ground pressure monitoring; microseismic signal; signal recognition; energy characteristic; fractal dimension

## 1. Introduction

### 1.1. Research Status

Wang et al. [1] used acoustic emission signals as the entry point to analyze the evolution process of limestone fractures and achieved good results. Chen et al. [2] revealed the mechanism of rock burst formation by analyzing the changes of microseismic signals during the formation of rock burst in coal/rock structural planes and also achieved good results. Tan et al. [3] analyzed the waveform characteristics of microseismic events through mathematical statistics, extracted the frequency domain, duration, and other related characteristics of signals, and constructed a classification and recognition model of microseismic events, which achieved good results. J.A. Vallejos [4] applied logistic regression analysis and computer neural networks to the identification of two types of event signals, and constructed a model using multiple parameter characteristics of the signal, which achieved good results. Dowla [5] used artificial neural network (ANN) to identify and classify different types of signals. The research shows that artificial neural networks (ANN) have more computational advantages than traditional identification methods. Eray Yildirim et al. [6] combined a feedforward neural network (FFNN), adaptive neuro-fuzzy inference system (ANFIS), and probabilistic neural network (PNN) to distinguish between mine earthquake and quarry blasting, and established a model with a high recognition rate.

Zeng Jianxiong [7] used multiple parameters in microseismic signal events as feature vectors and used support vector machines (SVM) to judge the category of microseismic signal events and achieved high accuracy. Wu Shengshen et al. [8] used a model based on genetic algorithm optimization to establish a BP neural network to analyze the stress changes during the installation of caissons with multiple influencing factors and achieved good results. It is proved that this method can analyze the complex situation of multiple factors.

In previous studies, most of them used a single-feature qualitative analysis (such as a single selection of certain aspects of the characteristics) or selected some waveform parameters as recognition features rather than using a unified analysis of multiple aspects of the characteristics. Although previous feature extraction methods can still perform preliminary identification, they are often not extensive. Many scholars who study microseismic signals mostly extract the characteristic parameters of microseismic signals by Fourier transform, wavelet transform, and wavelet packet decomposition and combine them with other pattern recognition methods to achieve waveform discrimination. However, the nonlinear and nonstationary characteristics of microseismic event signals are often ignored. The above several eigenvalue extraction methods are all aimed at the traditional stationary signal processing methods. When using the analysis method for processing stationary signals to process nonstationary signals, it may often cause errors.

Wang Guangjin et al. [9,10] and Lin Shuiquan et al. [11] used a comprehensive analysis of multiple parameters and established an analysis method for identification models in the analysis of tailings pond accumulation characteristics, tailings dam saturation lines, tailings flow evolution models, tailings dam breaks, and other complex situations containing multiple different parameters. Compared with the in-depth analysis of a certain parameter, comprehensive analyses of multiple parameters and the analysis of the correlation between parameters are carried out. Finally, the method of determining the analysis model can achieve better analysis results. In the analysis of slope stability and the establishment of an intelligent prediction system, Wang Guangjin et al. [12] fully considered the five parameters that affect the stability of the slope, comprehensively analyzed the five parameters, and combined the correlation between the five parameters. On this basis, the identification model was established and achieved good recognition results. It is proved that in the face of complex situations, comprehensive analysis of multiple parameters can achieve better results to a certain extent.

Therefore, this paper uses the analysis method of comprehensive analysis of multiple parameters to analyze and establish the microseismic signal recognition model and uses the Hilbert–Huang transform, which has a good effect on nonlinear and nonstationary signals, to process microseismic signals. Specifically, through time–frequency [13–28], energy [29–34], and fractal features [35–47], the characteristic values of rock fractures and rock blasting signals are extracted from three perspectives, and the differences between the two types of eigenvalues are compared, which provides a reference range for the manual identification and the necessary reference range for computerized automatic signal recognition.

### 1.2. Technical Route

In this paper, the comprehensive analysis method of multiple parameters is used to analyze the difference between rock blasting signals and rock fracture signals. The technology roadmap is shown in Figure 1. The specific analysis methods are as follows:

1. The differences between the maximum amplitude value and the signal duration of the collected original rock blasting signals and the rock fracture signals are directly analyzed.
2. Using Symlets8 wavelet, three layers of decomposition layers are selected to denoise the original microseismic signal. The Hilbert–Huang transform is used to process the microseismic signal after noise reduction, and the differences between the microseismic signal of rock blasting and the microseismic signals of rock fractures in the main frequency of the signal are analyzed.

3. The ensemble empirical simulation decomposition method is used to decompose the microseismic signal into 8 layers after noise reduction, and the differences between the energy proportion of the rock blasting microseismic signals and the rock fracture microseismic signals in terms of total energy proportion after different decompositions are analyzed.

4. The fractal box dimension method is used to analyze the differences in fractal box dimension distribution between rock blasting vibration signals and rock fracture microseismic signals after noise reduction.

5. According to the above analysis results, an automatic identification model of microseismic signals based on the BP neural network is established, and the model is trained and tested.

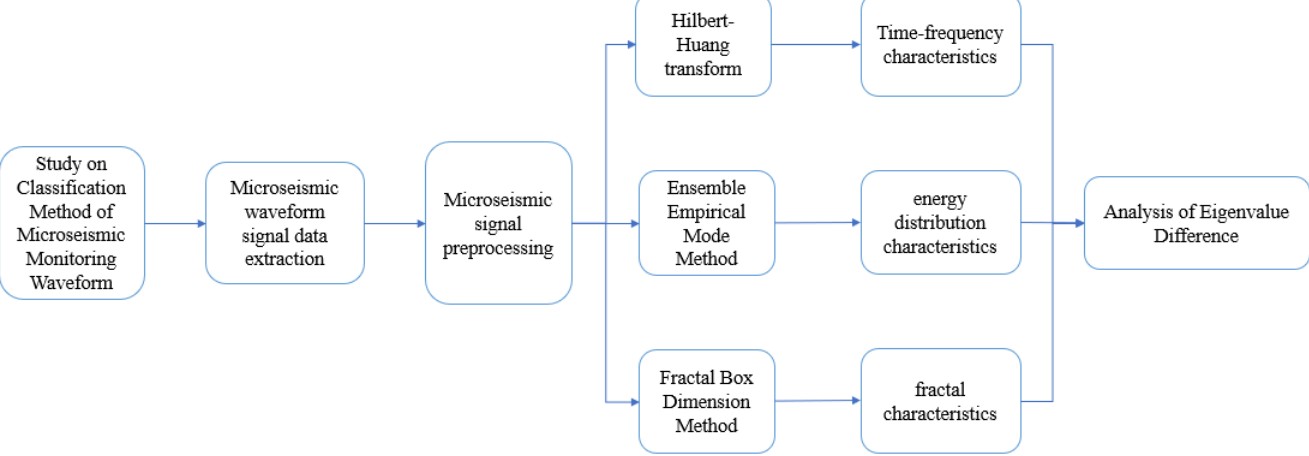

**Figure 1.** Technology roadmap.

## 2. Time–Frequency Characteristic Analysis of Microseismic Waveform Signal

### 2.1. Time Domain Analysis of Microseismic Signals

From the data collected by underground microseismic monitoring equipment, 50 groups of rock blasting and rock fracture signals were randomly extracted for statistical analysis. It can be observed from Figure 2a,b that there is a significant difference in the main duration ratio between the two types of signals: the duration of the rock fracture signals is mainly distributed in the range of 0–100 ms, accounting for 96% of the total sample, whereas the duration of the rock blasting signal is mainly distributed in intervals greater than 150 ms, accounting for 78% of the total samples. By comparing Figure 2a,b, it can be deduced that the durations of the rock rupture and rock blasting signals are both distributed in the range of 50–150 ms.

Figure 3a,b show the distribution and proportion of the maximum amplitude values in 50 groups of data of the two types of signals. By comparing Figure 3a,b, it can be deduced that the maximum amplitude of rock fracture signal is mainly distributed in the range of 0–500 MV, accounting for 94% of the total number of samples of rock fracture signal maximum amplitudes. The maximum amplitudes of the rock blasting signals are mainly distributed in the range above 3000 mv. However, 32% of these values of rock blasting signals are distributed in the range of 0–500 MV.

Therefore, using the signal duration and maximum amplitude value to distinguish the rock blasting and rock fracture signals can cause misjudgment. It is necessary to analyze the two types of signals to find a more accurate method to distinguish them.

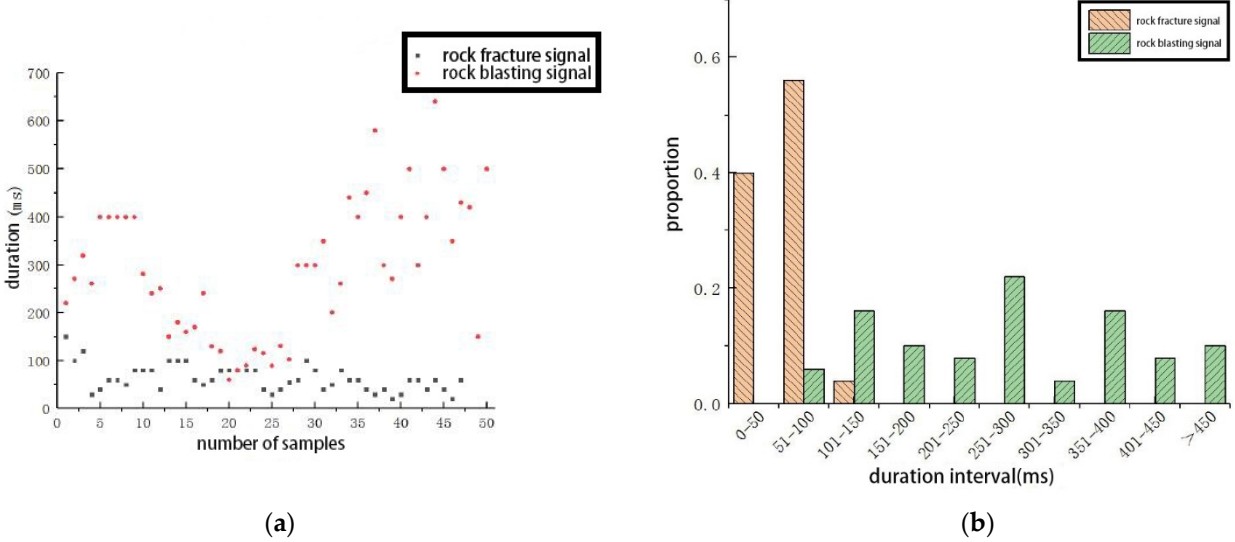

**Figure 2.** (**a**) Duration distribution of the two signals; (**b**) histogram of time ratio of two types of signals.

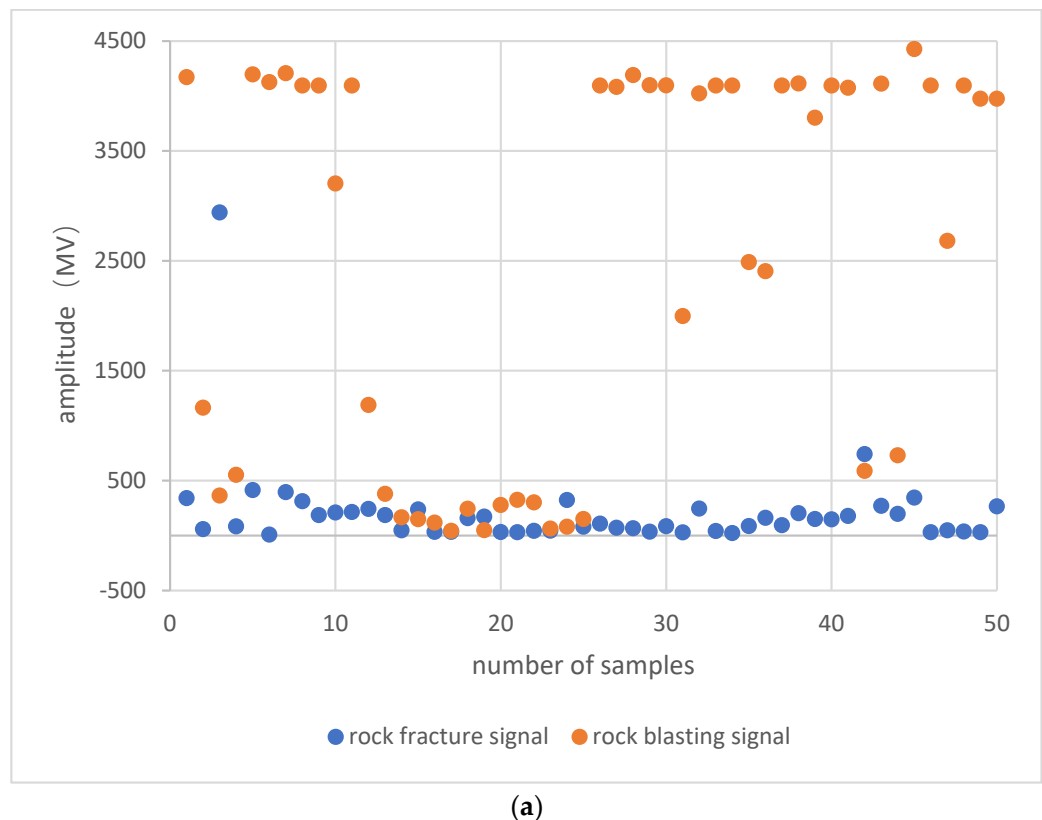

(**a**)

**Figure 3.** *Cont*.

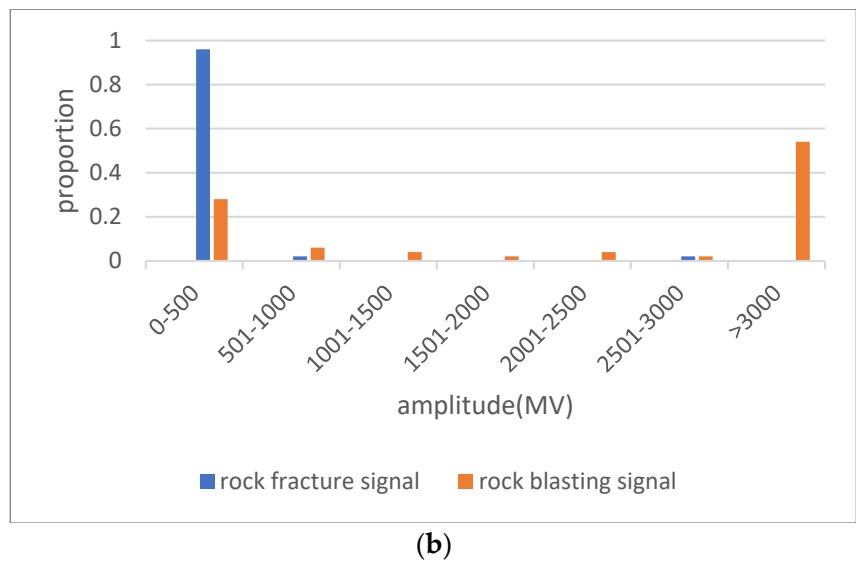

**(b)**

**Figure 3.** (**a**) Maximum amplitude distribution of two types of signals; (**b**) proportion diagram of maximum amplitude values of two types of signals.

### 2.2. Frequency Domain Analysis of Microseismic Signals

In this section, the frequency domain characteristics of rock fracture and rock blasting signals are analyzed, and the differences between these two types of signals in the frequency domain are studied. The typical signal waveform of the original rock fracture signal is shown in Figure 4, and the typical wave waveform of the original rock blasting signal is shown in Figure 5, but there are some original rock blasting signal waveforms similar to the original rock fracture signal waveforms shown in Figure 6. Therefore, it is necessary to process the original signal waveform in order to better distinguish the difference between the two types of signals in microseismic wave shape.

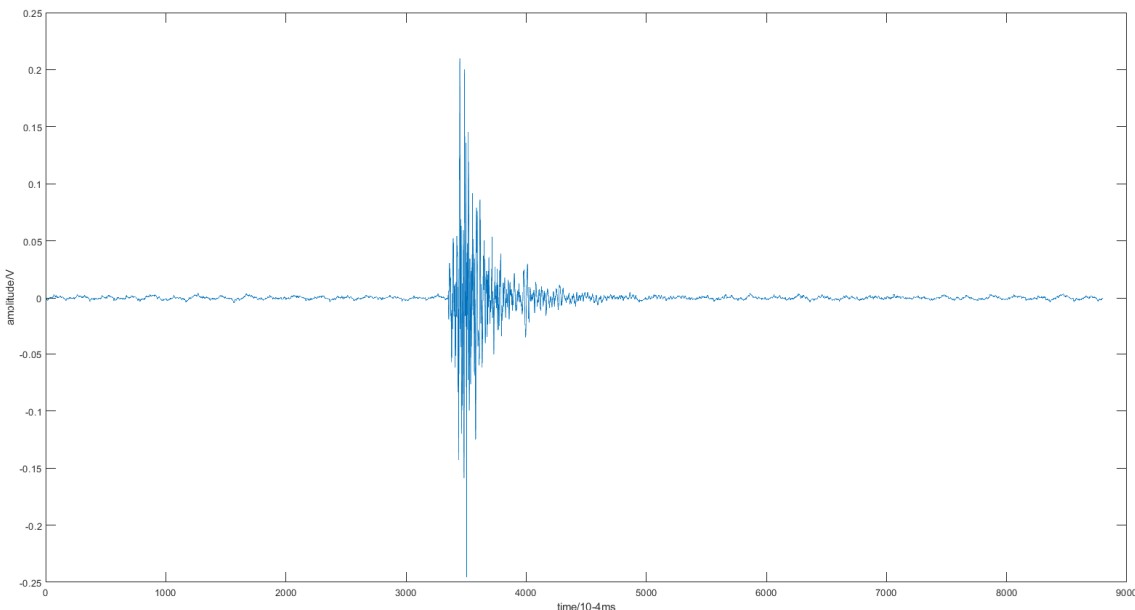

**Figure 4.** Original signal waveform of rock fractures from 40# sensor on 5 July 2019 (randomly selected rock fracture events).

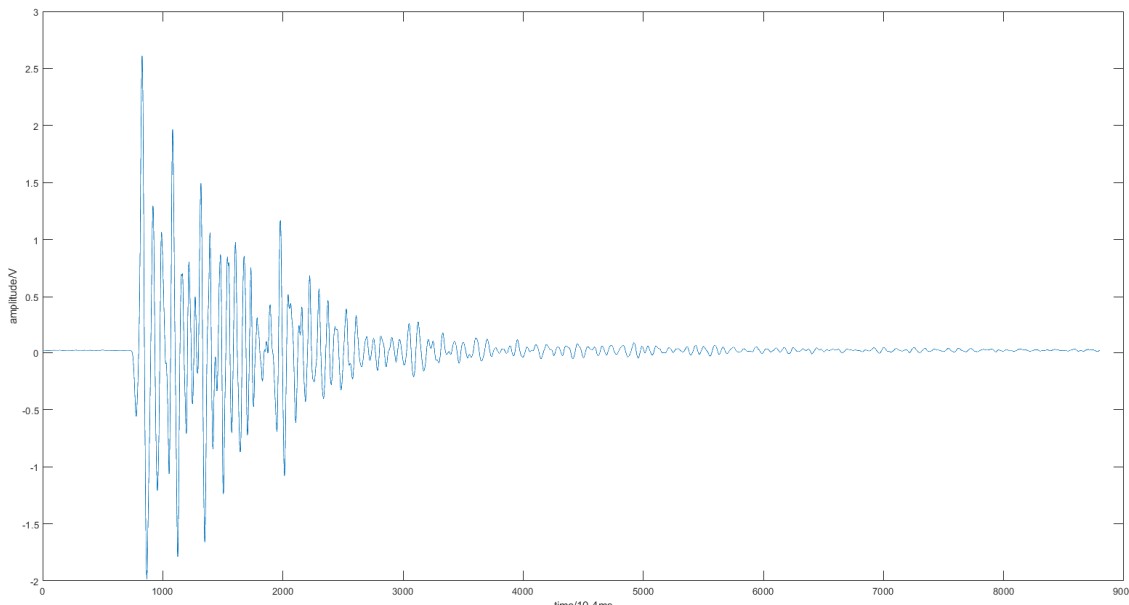

**Figure 5.** Original signal waveform of rock blasting at 33# sensor on 9 July 2019 (randomly selected rock blasting events).

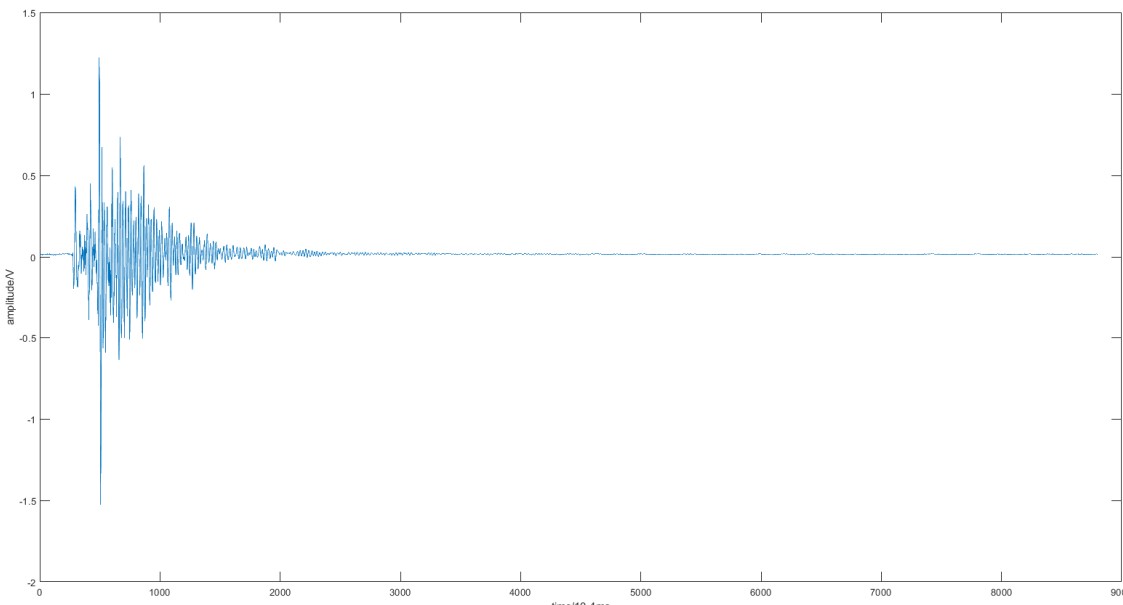

**Figure 6.** Waveform diagram of original signal of rock blasting by 33# sensor on 2 July 2019 (rock blasting event signal similar to rock fracture event signal).

To analyze the frequency domain of rock fracture time signals more accurately, it is necessary to first denoise them. Figure 7 shows a comparison between the frequency domain of the signal after noise reduction and the frequency domain of the original signal. Symlets8 wavelet was selected for noise reduction, and three layers were selected for the decomposition layer.

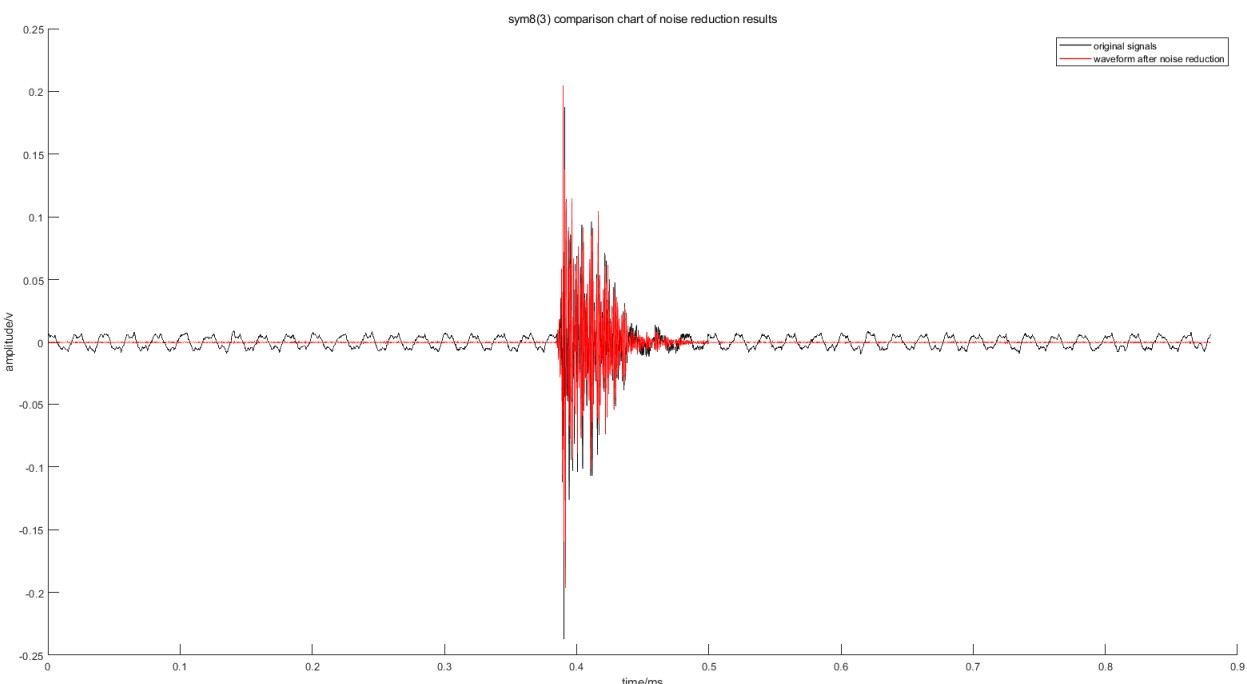

**Figure 7.** Contrast diagram of noise reduction results.

The Hilbert–Huang transform (HHT) is a signal analysis method [48–51] for nonlinear and nonstationary signals similar to microseismic waveform signals. This method has been applied to the processing and analysis of microseismic signals and has achieved good results [52–54]. Recently, Chen et al. [55] used HHT to analyze the acoustic emission waveform characteristics of rock under uniaxial loading and achieved good results.

The HHT was used to process the original and denoised signals to obtain the marginal spectrum of the original and denoised signals, respectively. The marginal spectra of the two signals are shown in Figure 8a,b, respectively. In the marginal spectrum, the corresponding interval of the signal peak value after processing was the main frequency value of the signal. By comparing Figure 8a,b, it can be deduced that after the noise reduction signal was processed, obtaining the signal with only one peak in the main spectrum became easier. Analysis of the original signal before denoising is necessary.

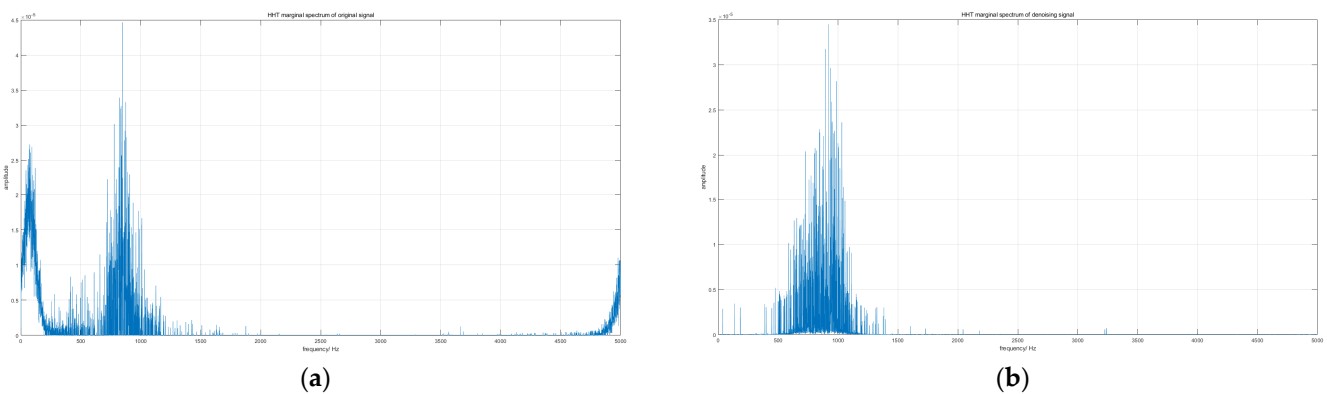

**Figure 8.** (**a**) The marginal spectrum of the original signal; (**b**) the marginal spectrum of the de-noised signal.

Therefore, for two rock blasting signals selected from the same sensor on different dates, the symlets8 wavelet base was first selected, the original signal was decomposed to three layers, and the HHT was used to process the denoised signal. The noise reduction results and marginal spectrum are shown in Figures 9 and 10, respectively. The results of a comparative analysis of Figures 9 and 10 are summarized as follows. The main frequency of the original signal was not affected by the wavelet-based denoising method; that is, after using HHT to convert the noise-reduced and non-noise-reduced signals, it was found that the main frequency of the signal is clearer after noise reduction on the premise of ensuring that the main frequency characteristics of the signal remain unchanged.

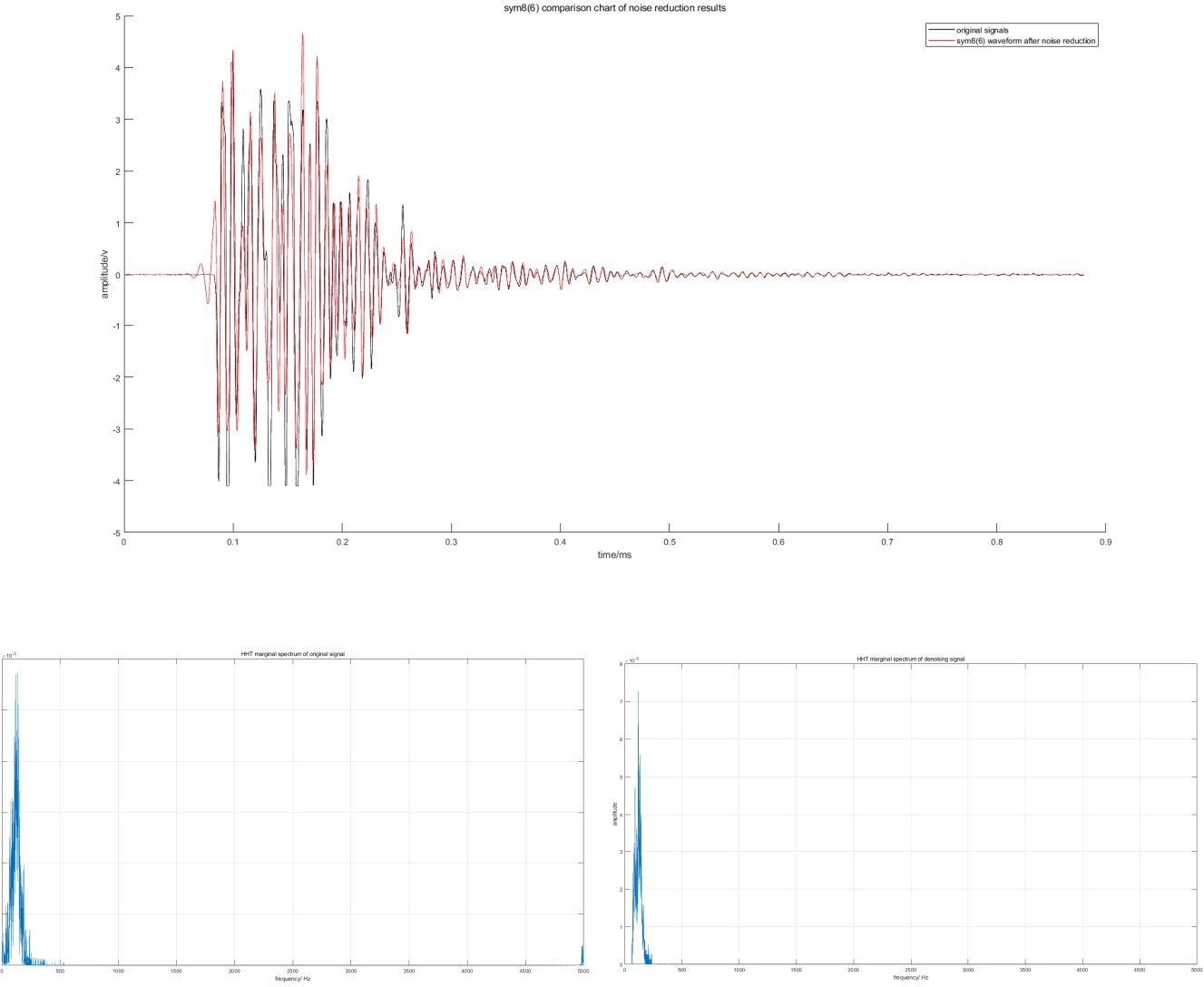

**Figure 9.** Contrast image of marginal spectrum after denoising of rock blasting signal of 33# sensor on 9 July.

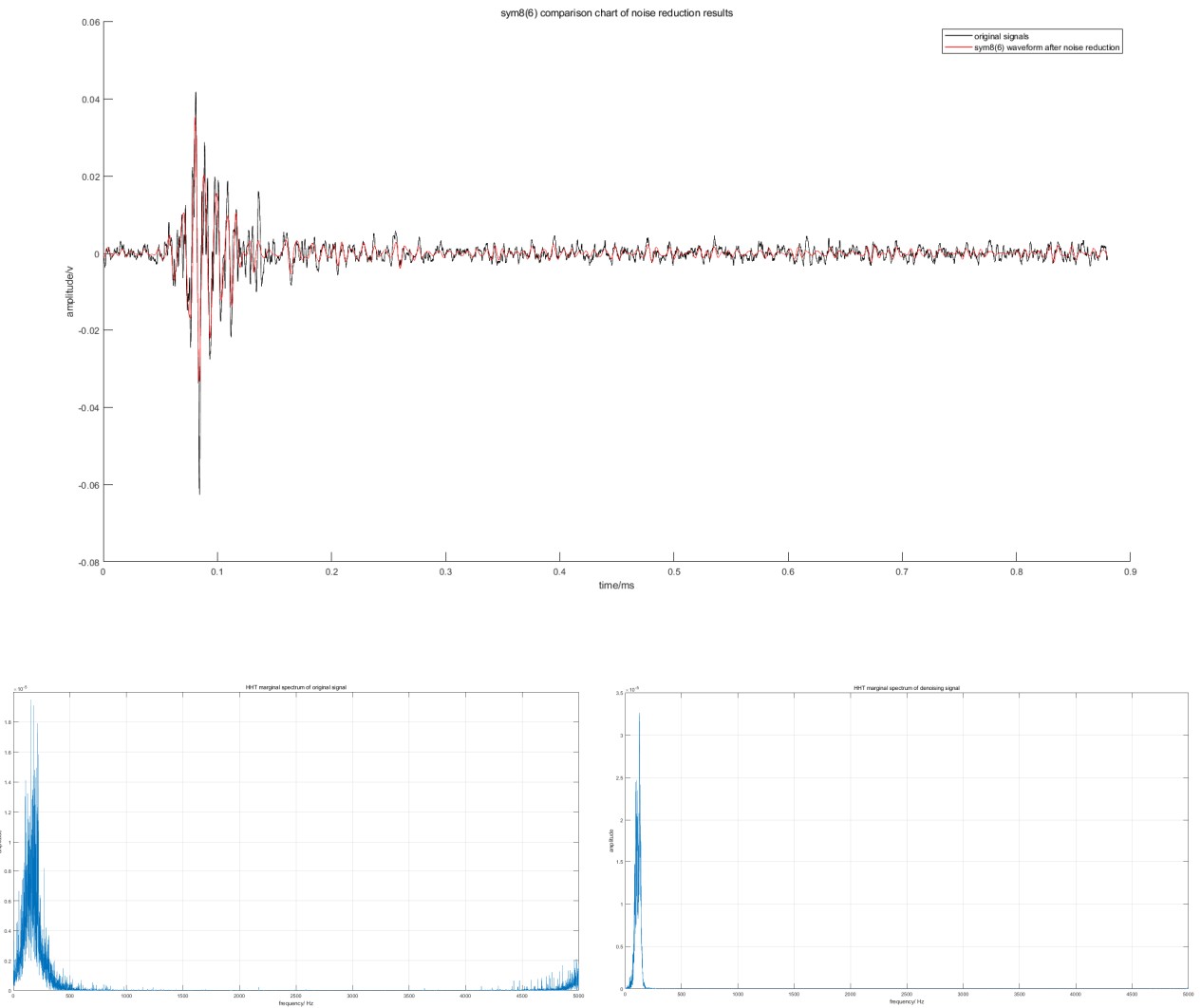

**Figure 10.** Contrast image of marginal spectrum after denoising of rock blasting signal of 33# sensor on 2 July.

To distinguish the two types of signals in the main frequency, HHT was applied to all 50 sets of burst event and blasting event signals. The main frequency was divided into five intervals. Table 1 is the statistical table of the interval where the main frequency of each event was located. Figure 11 shows the main frequency distribution of the two types of signals after statistics. By comparing Table 1 with Figure 11, it can be deduced that the main frequency of the rock blasting signal samples collected this time is mainly concentrated in the range of 0–500 Hz, and the main frequency of the rock fracture event samples collected this time is distributed in the range of more than 500 Hz.

**Table 1.** Statistical table of interval of main frequency of rupture and blasting events.

| Event Frequency (Hz) | 0–500 | 500–1000 | 1000–2000 | 2000–3000 | >3000 |
|---|---|---|---|---|---|
| Rock blasting | 47 | 2 | 1 | 0 | 0 |
| Proportion | 94% | 4% | 2% | 0 | 0 |
| Rock fracture | 0 | 14 | 15 | 1 | 20 |
| Proportion | 0 | 28% | 30% | 2% | 40% |

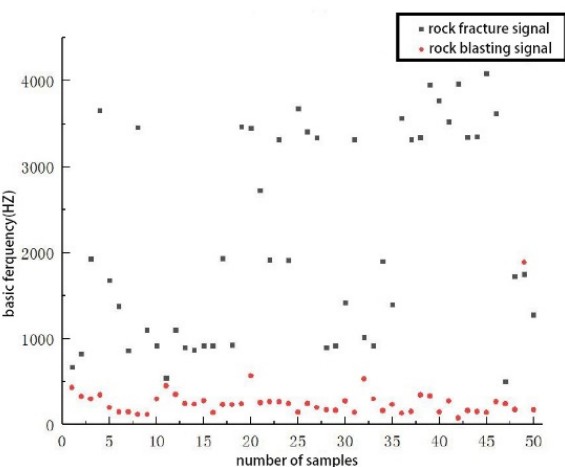

**Figure 11.** Dominant frequency distribution diagram of two types of signals.

From the above analysis, it can be observed that the main frequency of the signal after noise reduction can be used as the basis to accurately judge the signal of rock fracture and rock blasting events; however, misjudged individual signals still exist.

### 3. Energy Characteristic Analysis of Microseismic Waveform Signals

In this section, the band energy is used as the breakthrough point to study the characteristics of the frequency band energy of rock fractures and rock blasting signal waveforms. Furthermore, the ensemble empirical simulation decomposition (EEMD) method is used to analyze rock blasting and rock fracture signals.

First, the EEMD is used to process the selected signal. After processing, the set number of the intrinsic mode function (IMF) is obtained. According to the following equation, the energy value, $E_i$ (1, 2, 3 ... ... ), corresponding to each IMF is calculated.

$$E_i = \sum |IMF_i|^2 \tag{1}$$

$$E_{total} = \sum_{i=1}^{n} E_i \tag{2}$$

$$P_i = \frac{E_{total}}{E_i} \tag{3}$$

Through the three equations above, we can calculate the energy value contained in the intrinsic mode function of each frequency band and the ratio of the energy contained in a single intrinsic mode function to the total energy of the signal. $E_i$ denotes the energy of each intrinsic mode function, $E_{total}$ is the sum of energy contained in all intrinsic mode functions of the original signal, and $P_i$ represents the energy ratio in frequency domain of the signal.

The two types of signal data are first denoised using the wavelet threshold and then decomposed using the EEMD. The signal energy value corresponding to each interval decreases due to the increase in decomposition levels and eventually approaches zero as time approaches infinity. Therefore, to ensure that the results after decomposition are easy to analyze, eight-level decomposition is selected for the signal. The energy of eight IMF components and the ratio of the energy to the total energy are calculated. The difference in the main energy intervals of the two types of signals is analyzed for the purpose of signal discrimination.

EEMD is used to decompose the denoised signal into 8 layers. After decomposition, a total of 9 different spectrograms of IMF1–IMF8 and remainder are generated, which are arranged from high to low frequency. Corresponding to the IMF1–IMF8 and remainder, a total of nine corresponding frequency band energies are generated. The ratio of energy

of each frequency band to the total frequency band energy represents the proportion of different spectrograms in the denoised signal after decomposition.

An original rock fracture event signal is selected and processed by the wavelet threshold denoising method (Symlets8 wavelet was selected, and three layers were selected for the decomposition layer). The processing results are shown in Figure 12. By analyzing the results, it can be deduced that the other signals generated in the process of acquisition are approximately straight lines in the graph after processing, and the signal of the rock fracture event has not changed significantly, which indicates that the effect of the wavelet threshold method is significant. Therefore, after noise reduction, the signal energy value is more representative of the signal energy of the actual event.

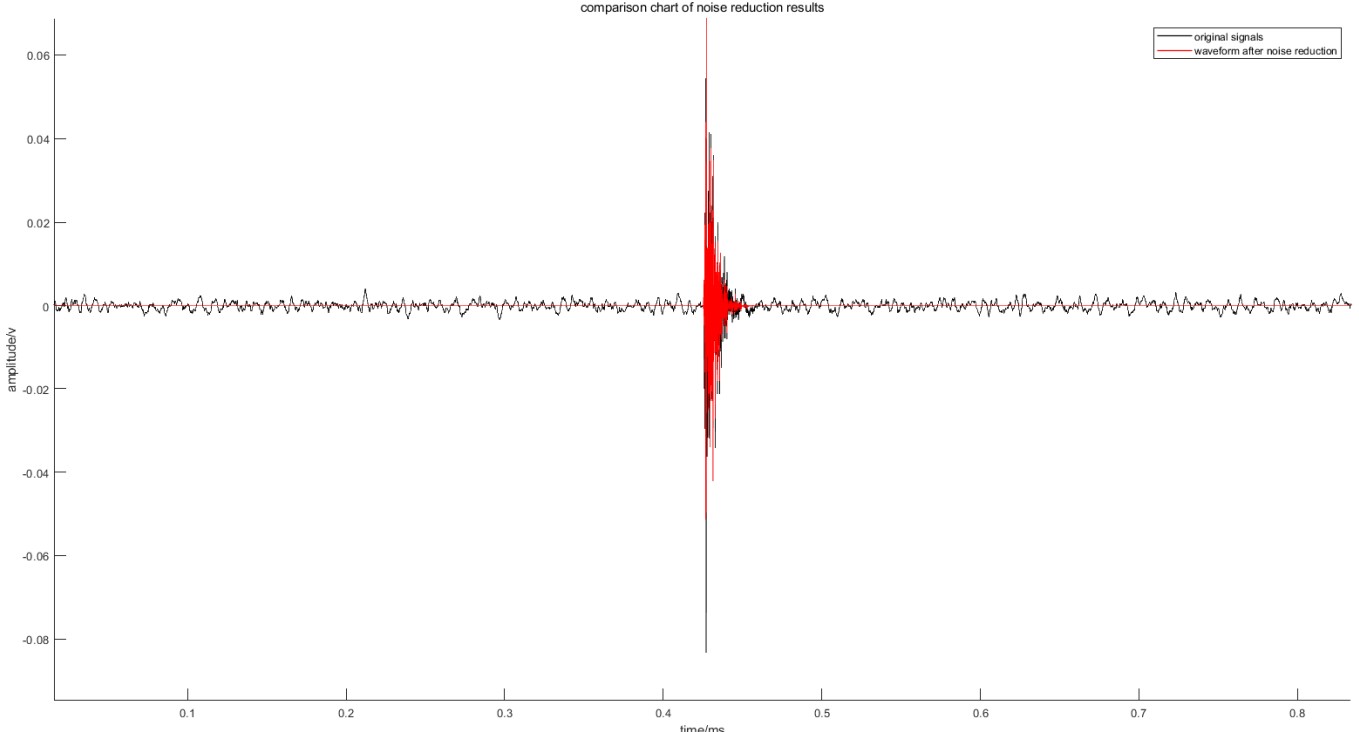

**Figure 12.** Contrast diagram of rock fracture signal denoising.

The signal of a rock fracture event after noise reduction is selected and processed by the EEMD. The spectrum is shown in Figure 13. By analyzing the spectrum, it can be deduced that the waveform displayed in the IMF1 frequency spectrum exhibits the highest resemblance to that after noise reduction; the spectrum of the other frequency bands indicates that the similarity between the waveform and denoised signal gradually decreases until it is almost negligible.

A raw rock blasting signal is selected and denoised using the wavelet threshold method. The processing results are shown in Figure 14.

By analyzing the results, it can be deduced that the other signals generated in the acquisition process are almost straight lines in the graph after EEMD processing, and the curve of the rock fracture event signal is smoother after noise reduction, which indicates that the effect of the wavelet threshold method is significant.

The signal of the rock fracture event after noise reduction is selected and processed by the EEMD. The frequency spectrum is shown in Figure 15.

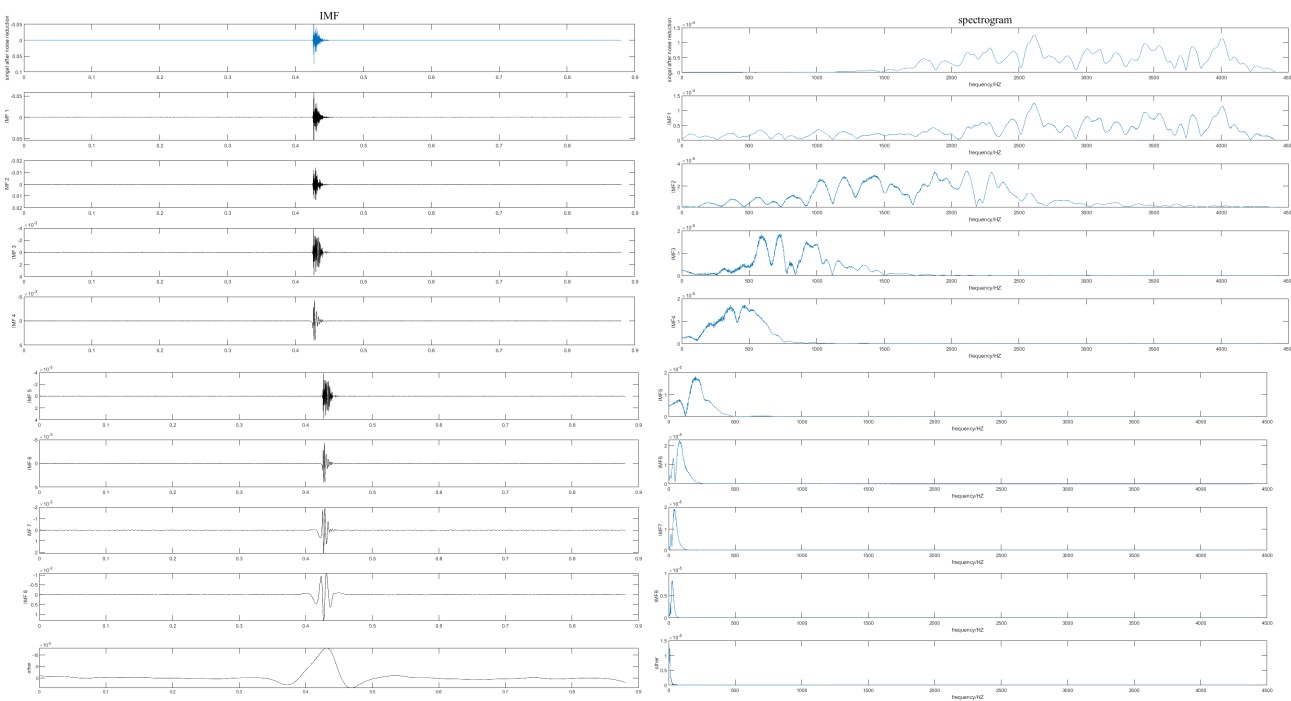

**Figure 13.** EEMD decomposition diagram and spectrum of rock fracture signal.

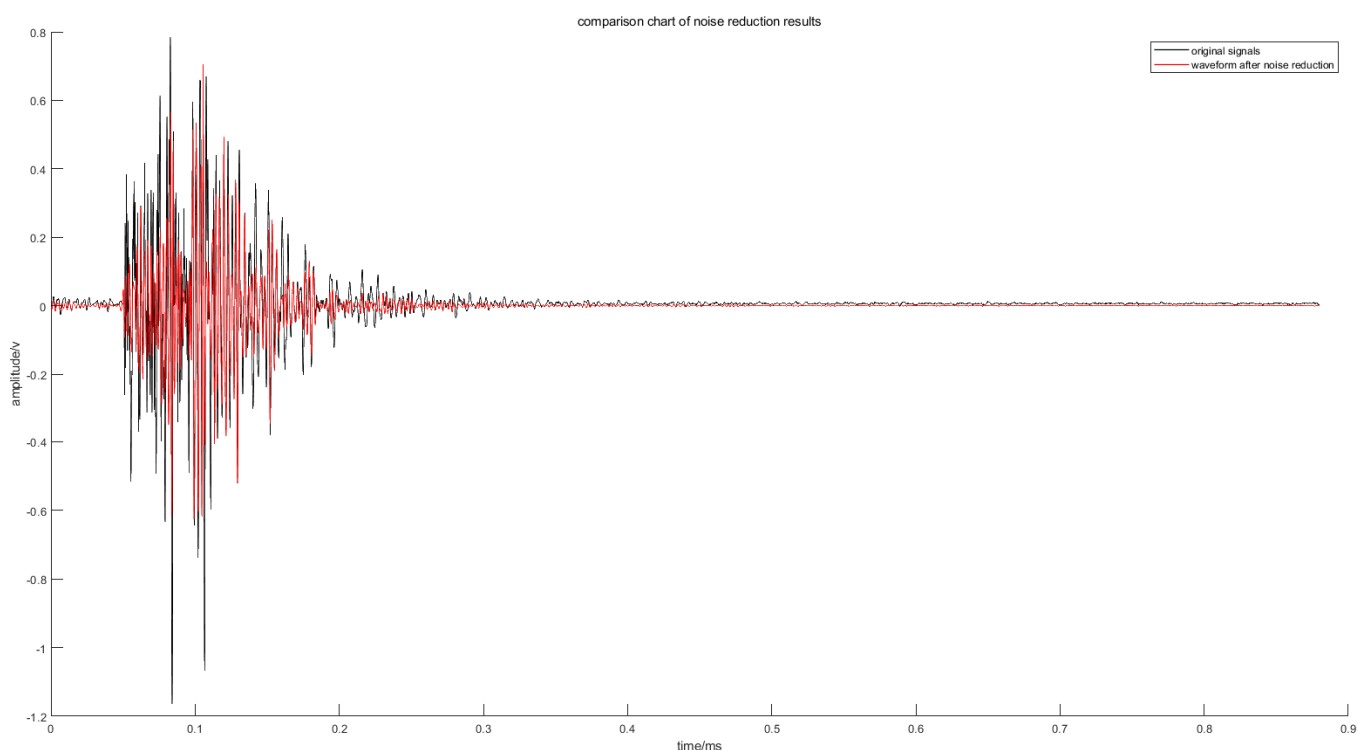

**Figure 14.** Contrast diagram of rock blasting signal noise reduction.

To compare and analyze the difference between rock blasting and rock fracture signals, eight-layer decomposition is selected as the same decomposition level of the two types of signals. The energy of each decomposition level of the two signals after decomposition is shown in Figure 16a,b.

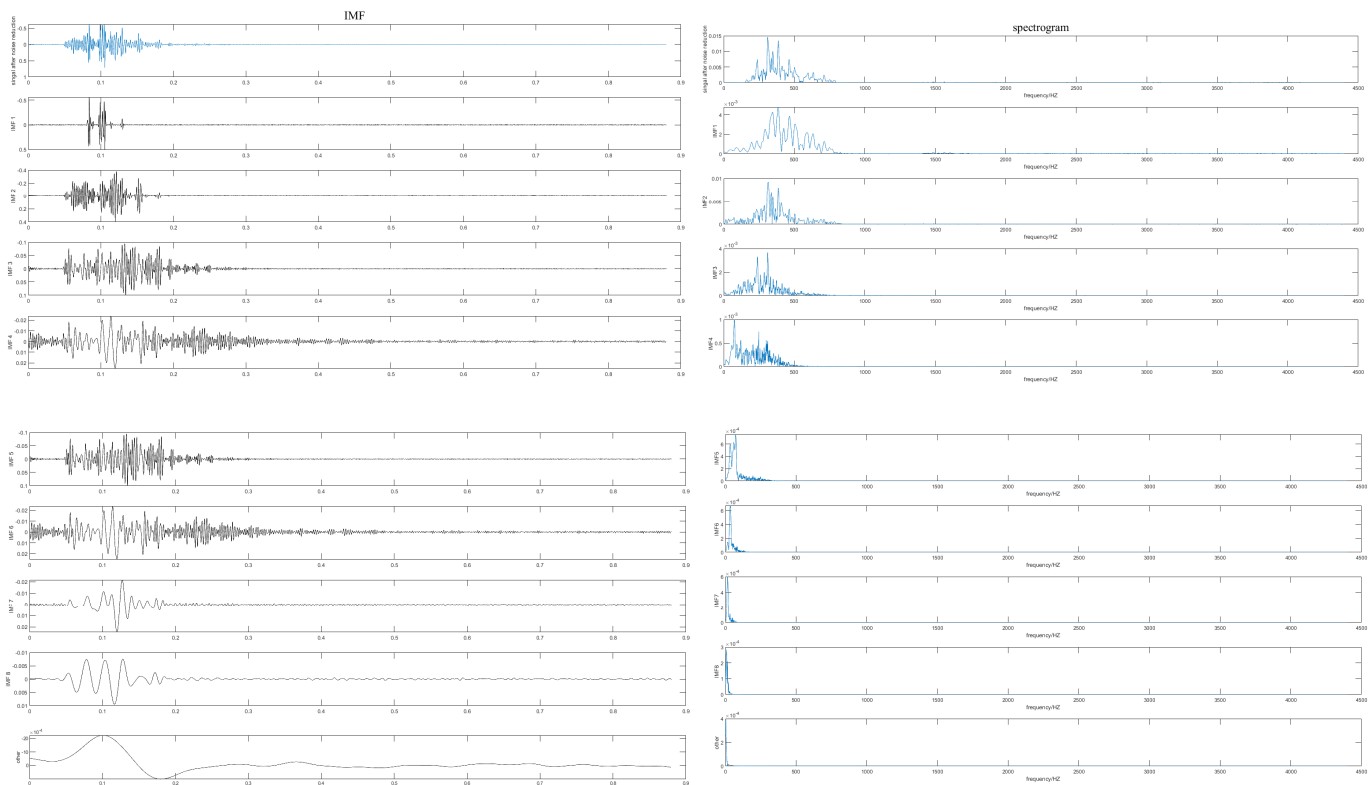

**Figure 15.** EEMD decomposition diagram and spectrum of rock blasting signal.

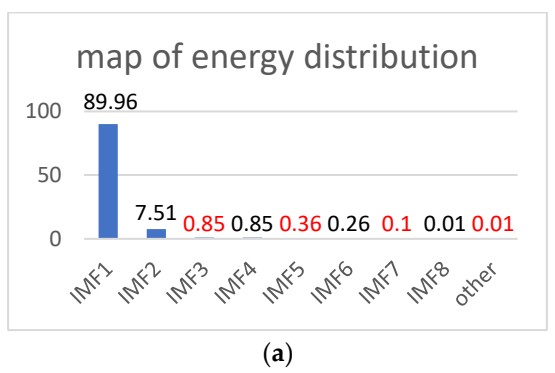

(**a**)

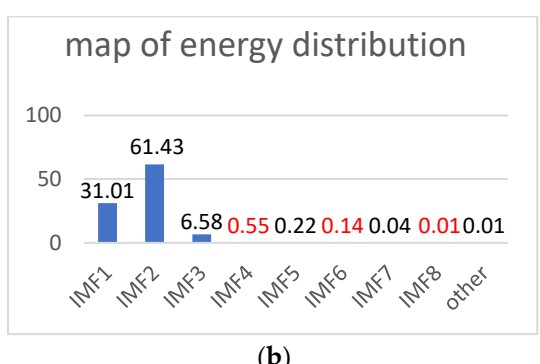

(**b**)

**Figure 16.** (**a**) Energy distribution diagram of each frequency band of rock fracture signal; (**b**) energy distribution diagram of rock blasting signal in each frequency band.

From Figure 16a, it can be observed that the signal energy of the rock fracture signal collected by the sensor is mainly concentrated in the IMF 1 frequency band, which accounts for 89.96% of the total energy.

From Figure 16b, it can be observed that the signal energy of the blasting signal collected by the sensor is mainly concentrated in the IMF1 and IMF2 bands, which account for 31.01% and 61.43% of the total energy, respectively.

For 50 groups of the two types of signals extracted using the microseismic monitoring equipment, the wavelet threshold method is used for noise reduction, and EEMD is subsequently used for eight-layer decomposition of the denoised signal. The results of the rock fracture signal decomposition are shown in Figure 17. The results of the rock blasting signal decomposition are shown in Figure 18.

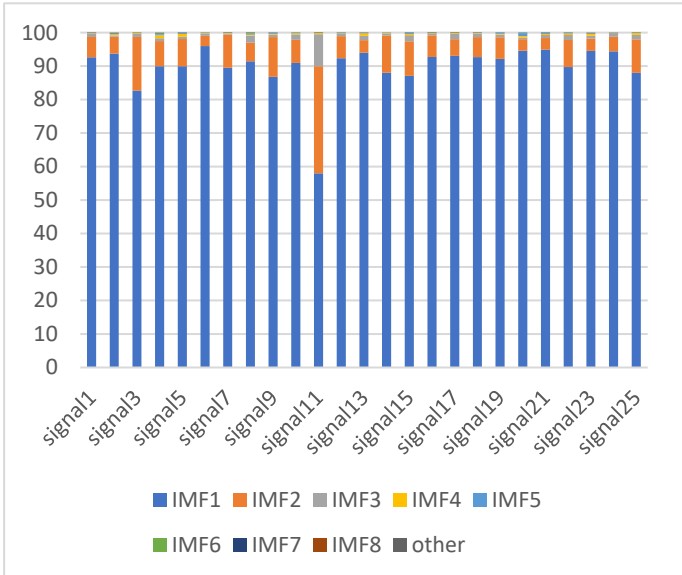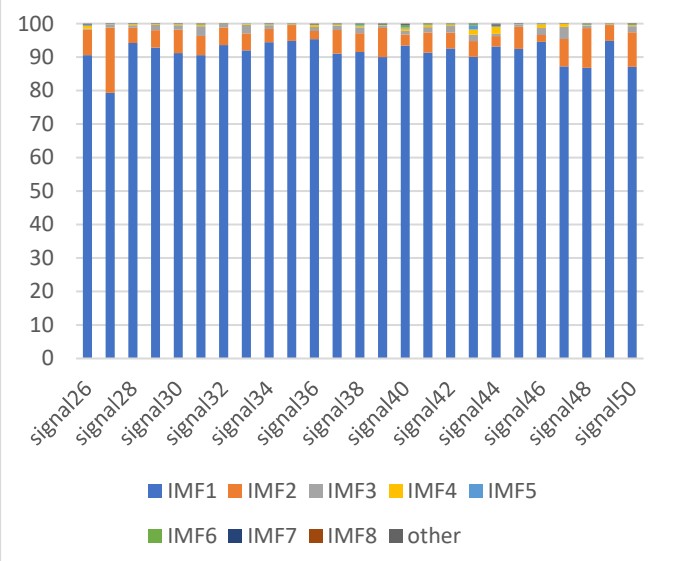

**Figure 17.** Histogram of energy percentage of each frequency band in total energy of rupture signal.

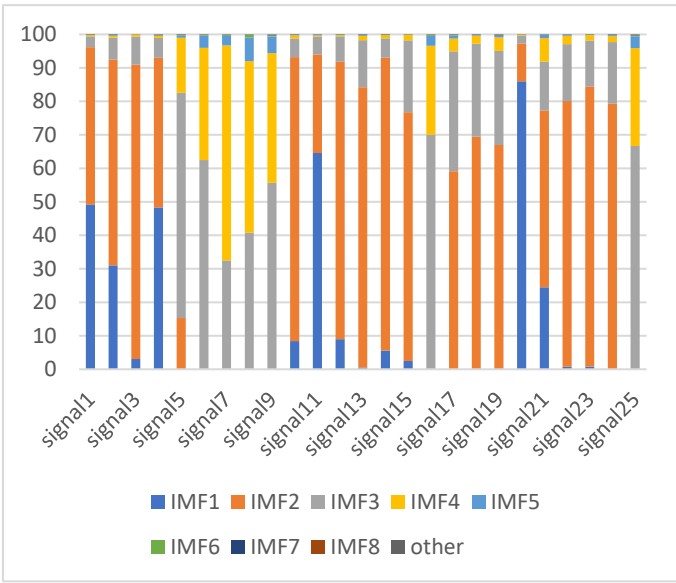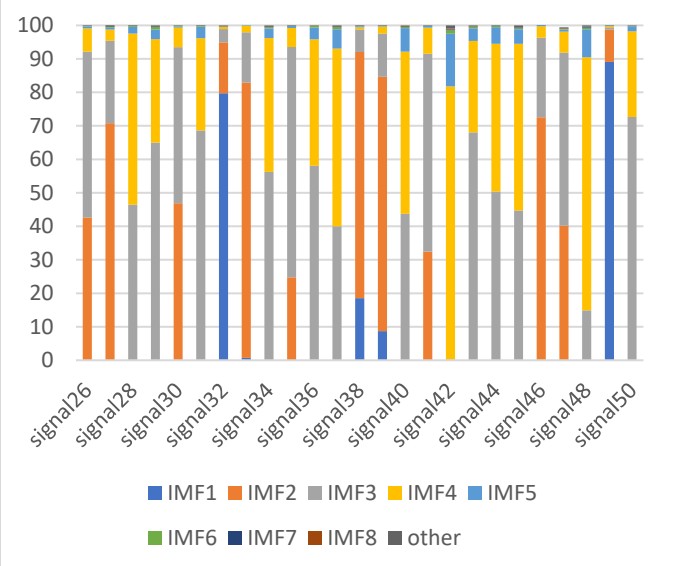

**Figure 18.** Histogram of energy percentage of each frequency band in total energy of blasting signal.

The following conclusions are drawn by comparing Figures 17 and 18.

1. Most of the energy of rock fracture signal is concentrated in the IMF1 frequency band, and the distribution energy in the IMF1 frequency band accounts for more than the total energy of the other frequency bands.
2. In the rock blasting signals, most of the energy is concentrated in the IMF 2, IMF 3, and IMF 4 bands; however, a few signal events were mainly observed in the IMF 1 energy band.
3. The signal category can be preliminarily judged by verifying whether the main energy is concentrated in the IMF 1 frequency band.

## 4. Fractal Feature Analysis of Microseismic Waveform Signals

Wang et al. [56] used mathematical methods to analyze the spatial distribution of microcracks in the broken state and achieved good results. Combined with the waveform characteristics of microseismic signals, the fractal function is selected to analyze the mathematical characteristics of microseismic waveforms.

In the 1980s, some researchers presented the multifractal theory using generalized-dimension and multifractal-spectrum mathematical methods to study a fractal object [45]. According to the definition of the fractal dimension, to ensure that a signal satisfies the fractal condition, it is only necessary to judge whether the signal has a scale-free region [46]. Xie et al. [47] conducted a vibration test research on a waveform. The fractal dimension can be used to classify different types of signals.

Among the many methods used to solve fractal dimensions, the fractal box dimension method is one of the most widely used [48]. Various Chinese researchers have conducted extensive research on microseismic waveforms and blasting vibration signals using fractal box dimensions [26–31,57–60] and achieved good results.

### 4.1. Fractal Box Dimension Calculation Method

Different microseismic signal waveforms are regarded as a rectangular grid composed of different lengths and widths. The transverse scale $\delta_1$ representing the rectangular length is determined by the sampling time of the signal, and the longitudinal scale $\delta_2$ representing the rectangular width is determined by the amplitude of the signal vibration. Assuming that the time history curve of each microseismic signal is represented by the mathematical calculation formula $L \in R^2$, $R \times R$ is divided into grids as small as possible to intersect with the time-history curve L. Assuming that the number of grids intersecting with the time-history curve L is $N_{K\delta_1}$ (or $N_{K\delta_2}$), the fractal box dimension under the rectangular box coverage is [61]:

$$D_{\delta_1 \times \delta_2} = \lim_{\substack{\delta_1 \to 0 \\ \delta_2 \to 0}} \frac{\log N_{K\delta_i}}{-\log k\delta_i} (i = 1 \ or \ 2) \tag{4}$$

Fractal box dimension is commonly used to express fractal dimension [61]. For microseismic waveform signals, it refers to the existence of a range in which $-\log k\delta_i$ and $\log N_{K\delta_i}$ have almost constant slopes, and the $(\delta_1, \delta_2)$ interval is called the scale-free interval of the waveform signal. For the same type of waveform signal, in theory, the fractal box dimension values in its scale-free range are close to each other. The fractal box dimension $D_{\delta_1 \times \delta_2}$ is obtained by calculating the slope of the double logarithmic curve for the scale-free interval $(-\log k\delta_i, \log N_{K\delta_i})$ $(i = 1 \ or \ 2)$.

The specific solving steps are as follows:

(1) Select a reasonable scale (that is, the width of the grid $\delta$) and divide the plane with waveform signal into equidistant grids. The abscissa (time) of the plane is divided into $Q_T$ equidistant grids, and the ordinate (amplitude) is divided into $Q_A$ equidistant grids.

$$Q_T = [T \div \delta] + \varphi(rem(T, \delta)) \tag{5}$$

$$Q_A = [A \div \delta] + \varphi(rem(A, \delta)) \tag{6}$$

When $x > 0$, $\varphi = 1$; when $x = 0$, $\varphi = 0$; $rem(u, v)$ is the remainder of the division of $u$ and $v$, $[T \div \delta]$ and $[A \div \delta]$ are integers.

(2) Find the number of fractal box dimension $N_x$ in each abscissa unit length interval.

$$N_x = [A_{max} \div \delta] - [A_{min} \div \delta] + \varphi(rem(A_{max}, \delta)) + \varphi(rem(A_{min}, \delta)) \tag{7}$$

(3) Find the total box dimension $N_{K\delta}$ after the whole waveform signal intersects with the grid.

$$N_{K\delta} = \sum_{x=1}^{Q_T} N_x \tag{8}$$

(4) Select different scales (grid width) δ, repeat the above (1)~(3) to obtain the corresponding $N_{K\delta_i}$, and calculate the slope $D_{\delta_1 \times \delta_2}$ and constant $b$, where $\log k\delta_i$ and $\log N_{K\delta_i}$ satisfy the following linear regression equation:

$$\log N_{K\delta_i} = D_{\delta_1 \times \delta_2} \log k\delta_i + b \tag{9}$$

In the formula, $D_{\delta_1 \times \delta_2}$ represents the calculated fractal box dimension, k represents the magnification of the rectangular box, and $b$ is a constant. Figure 19 is a schematic diagram of fractal analysis of microseismic waveform signals [44].

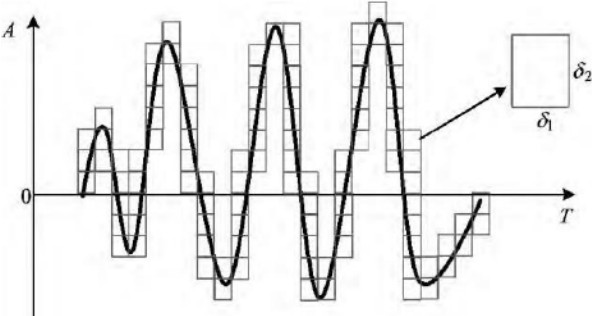

**Figure 19.** Fractal characteristics of microseismic waveform.

Scale-free interval:

For rock fracture signals and rock blasting signals, which cannot be described by mathematical functions, the signal waveform meets the conditions of fractal theory only in the scale-free interval of microseismic signals, and the fractal dimension can be calculated [42]. Therefore, it is necessary to determine the scale-free interval of microseismic waveform.

The value range of δ is mainly related to the sampling time $T$ and amplitude A of the microseismic signal. The sampling time interval $\Delta t$ of the sensor for mine acquisition signal is 1/10,000 s. The rectangular box width $\Delta w = k\delta_1$ needs to meet the following two conditions: the first condition is that it be not less than the time interval $\Delta t$, and the second condition is that it be not more than half of the sampling time T/2. The height of the rectangular box $\Delta h = k\delta_2$ also needs to meet two conditions: the first is that its value must be greater than 0, and the second is that its value cannot exceed the maximum or minimum amplitude value ($A_{max}$ or $A_{min}$) in the microseismic signal waveform. The minimum value of $k$ is 1, and the maximum value is $k = lg_2(T/\Delta t)$. Therefore, the width $\Delta w$ of the rectangular box and the height $\Delta h$ of the rectangular box can indicate that the microseismic signal has a scale-free zone as long as the following two mathematical formulas are satisfied.

$$\Delta t \leq \Delta w = 2^{k-1}\Delta t < \frac{T}{2}(1 \leq k \leq lg_2\left(\frac{T}{\Delta t}\right)) \tag{10}$$

$$\Delta h = |A_{max} - A_{min}| \times \Delta w/t \tag{11}$$

### 4.2. Fractal Box Dimension Analysis

Three groups of signals were randomly selected from 50 rock fracture signals and 50 rock blasting signals. The selected signals were processed by fractal processing, and the corresponding linear regression equations were obtained. According to the linear regression equations, the fractal box dimensions ($D_{\delta_1 \times \delta_2}$) and constant ($b$) were obtained. MATLAB software was used to calculate the parameters according to the above method, and the results are shown in Figures 20–25.

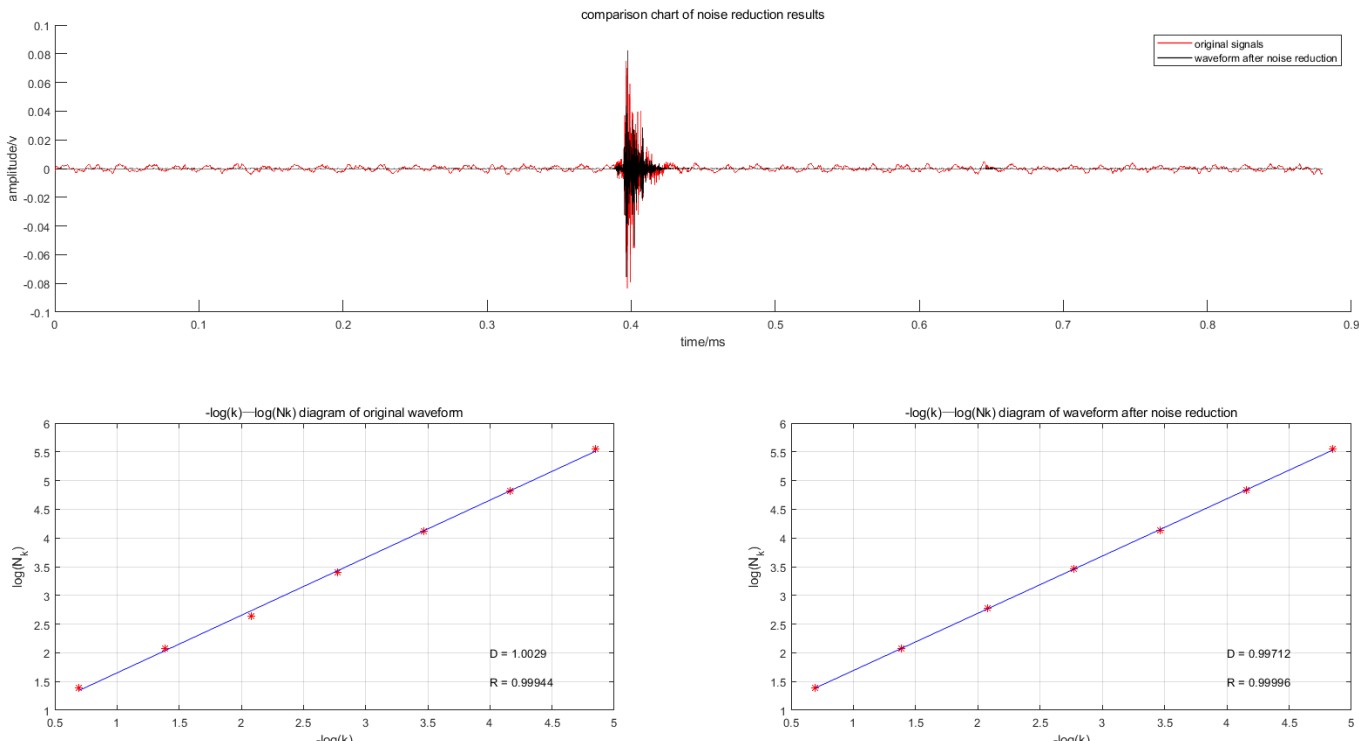

**Figure 20.** Fractal box dimension of no. 1 rock fracture event.

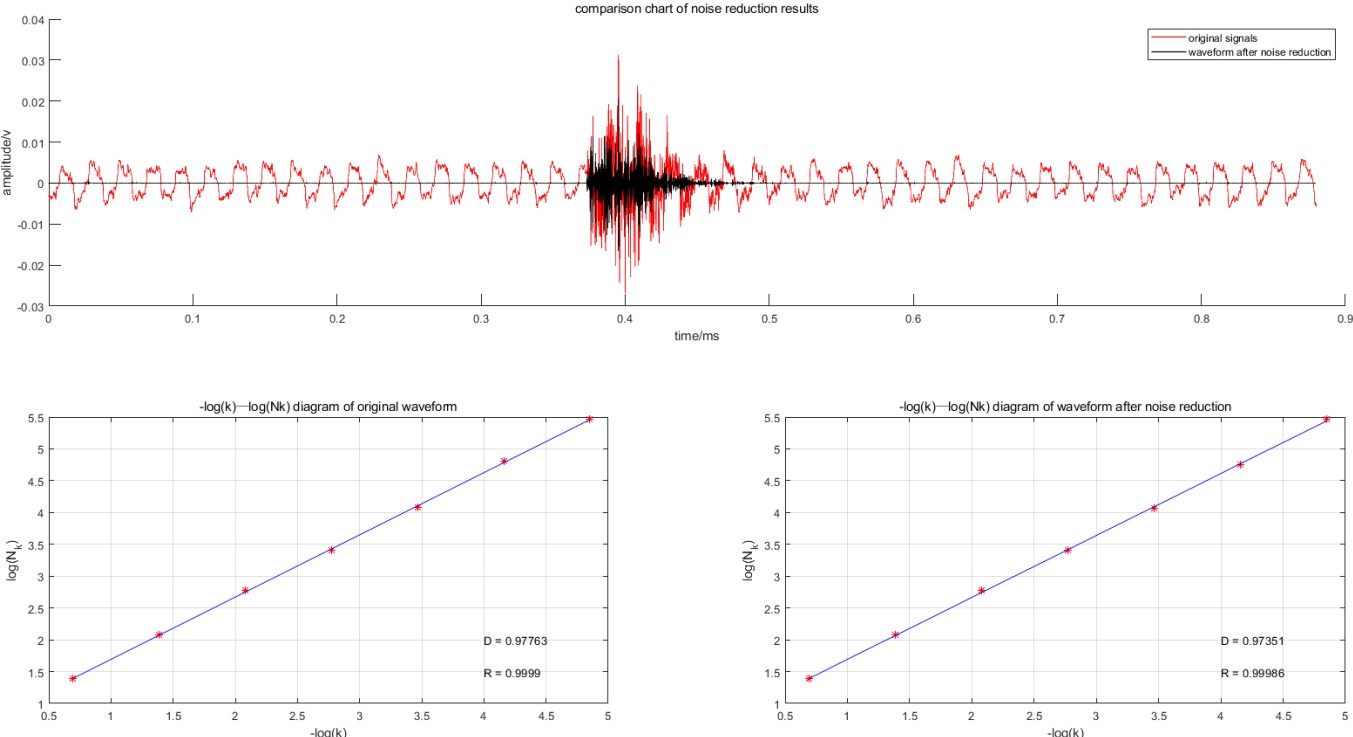

**Figure 21.** Fractal box dimension of no. 2 rock fracture event.

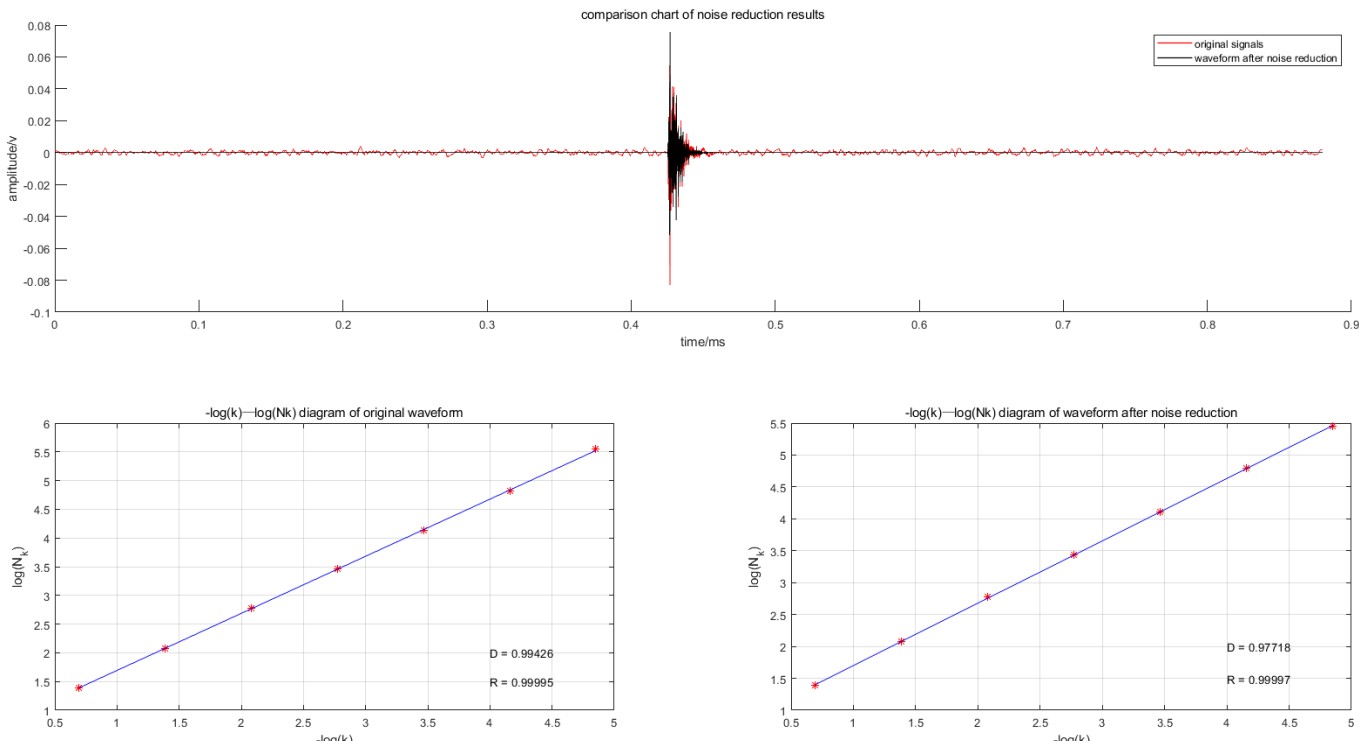

**Figure 22.** Fractal box dimension of no. 3 rock fracture event.

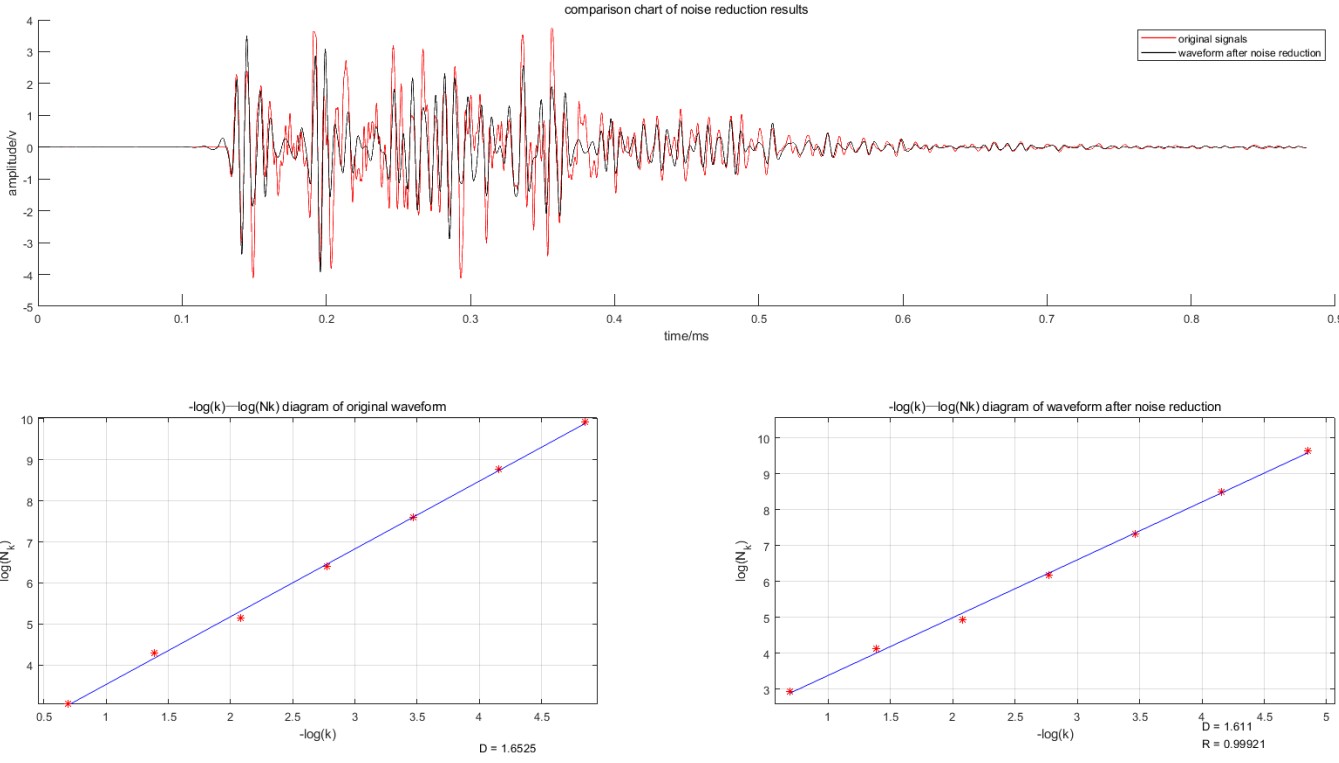

**Figure 23.** Fractal box dimension of no. 1 rock blasting event.

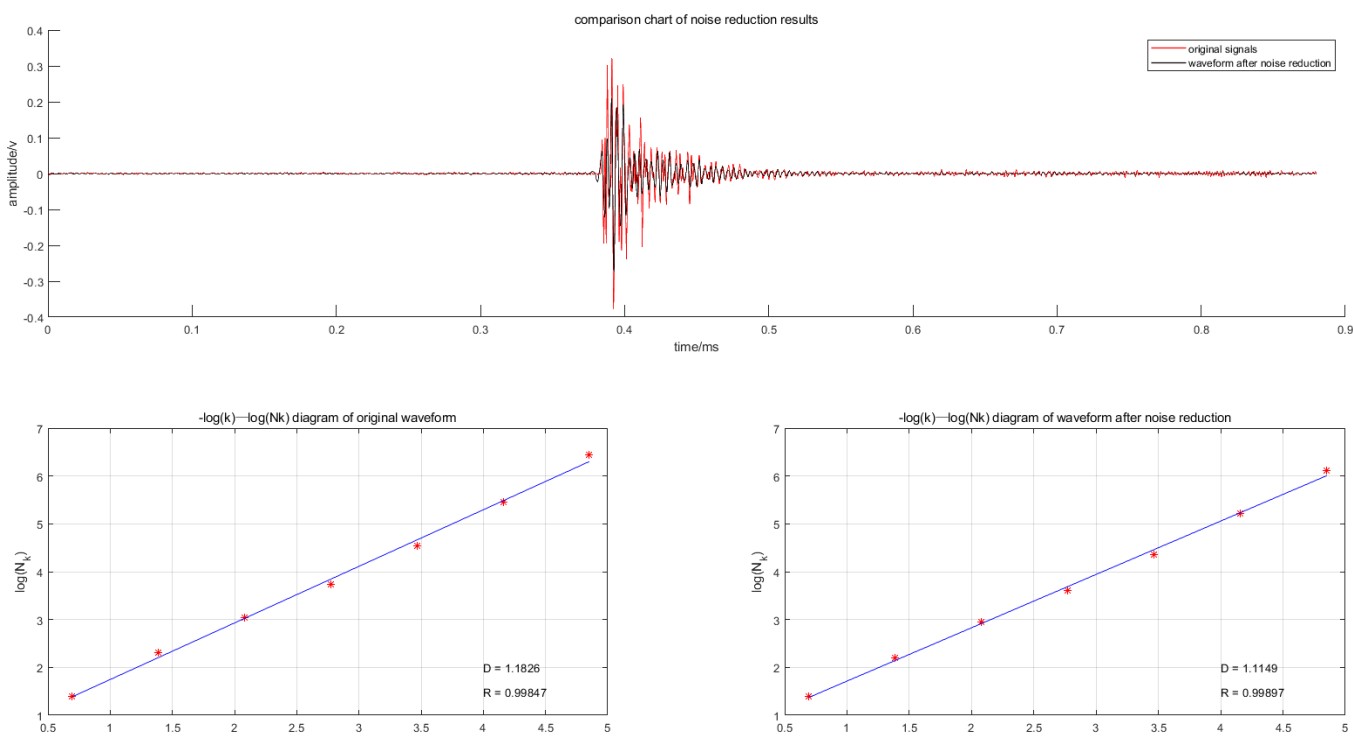

**Figure 24.** Fractal box dimension of no. 2 rock blasting event.

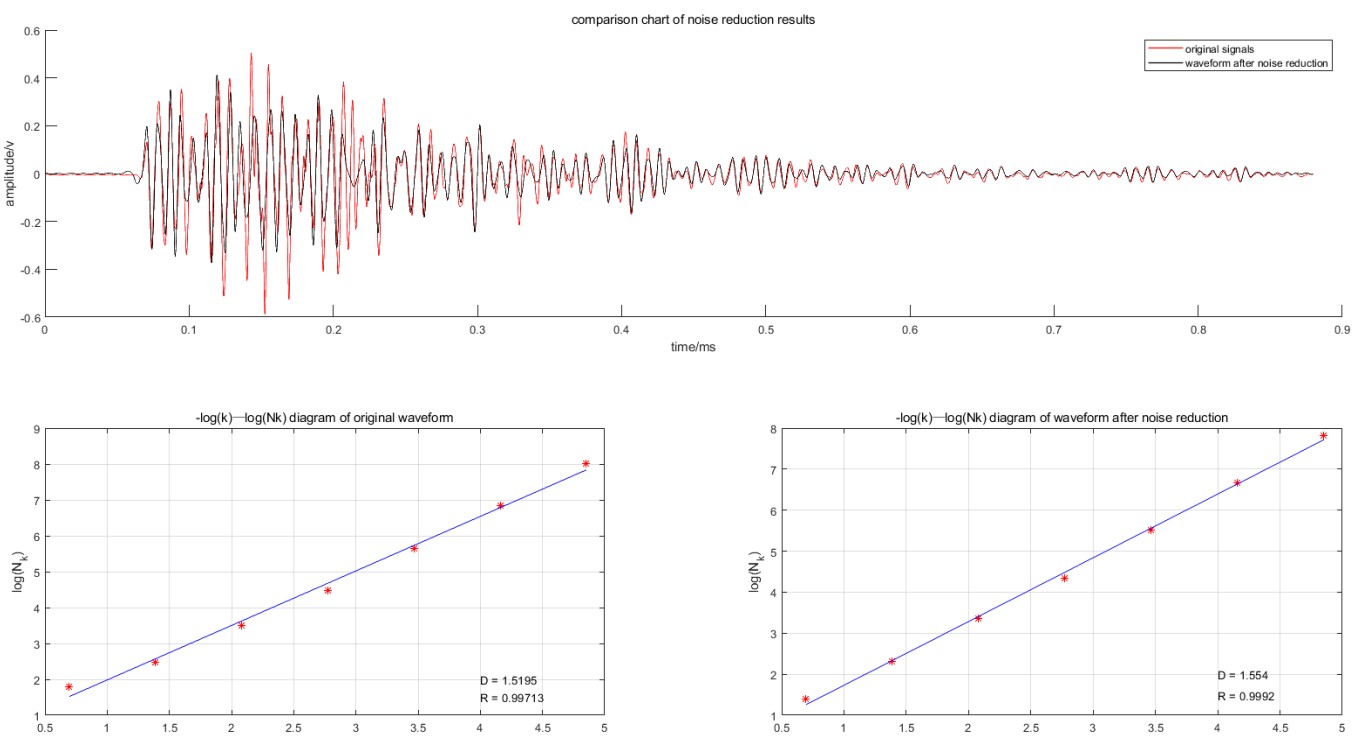

**Figure 25.** Fractal box dimension of no. 3 rock blasting event.

By comparing Figures 20–25 for the rock blasting and the rock fracture signals, it can be observed that the fractal box dimension of the signal without noise reduction is different from that of the signal with noise reduction; however, the difference is small. In addition, it was confirmed that the noise signal collected with the target signal demonstrates a

significant influence on the characteristic value of the signal. Therefore, it is necessary to denoise the original signal to reduce the impact of noise in the subsequent analysis.

Through a comparative analysis of Figures 20–25, it was observed that the fractal box dimension of the rock fracture signal after noise reduction was significantly different from that of the rock blasting signal after noise reduction. To clarify the difference in the points of the fractal box dimensions of the two types of signals, the fractal box dimensions of the selected rock fracture and rock blasting signals were calculated. The calculation results are shown in Figure 26.

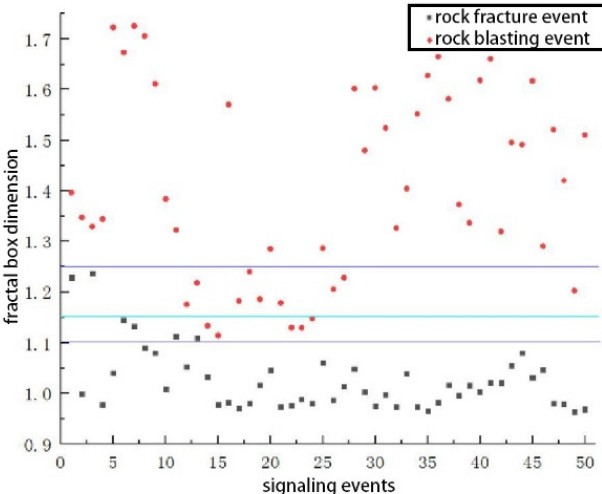

**Figure 26.** Fractal box dimension scatter distribution diagram.

According to an analysis of Figure 26, it was observed that the fractal box dimension of nearly 90% of the rock fracture signals was below 1.1. The fractal box dimension of nearly 70% of the rock blasting signals was more than 1.25. In addition, there was an uncertain interval of the fractal box dimension (i.e., interval of 1.1–1.25). After a comprehensive analysis, it was suggested that the fractal box dimension 1.15 should be selected as the fractal box dimension dividing line to distinguish the two types of signals. The accuracy of the rock fracture signal event judgment reached 88.89%, and the accuracy of the rock blasting event judgment reached 95.74%.

## 5. Comprehensive Discrimination of Microseismic Waveform Signals

It was observed that the microseismic signals of rock fracture and rock blasting events are not distinguishable in terms of signal duration and maximum amplitude.

There are some differences between the microseismic signals of rock rupture and the rock blasting events in the main frequency range, which can be obtained through the energy proportion characteristics of each layer after the eight-layer decomposition by the EEMD and fractal dimension.

In order to improve the accuracy of signal recognition, the choice BP neural network model, which can be used to deal with the classification of nonlinear signals such as microseismic waveform signals, is widely used and relatively mature to comprehensively discriminate microseismic signals. The operation process of BP neural network is shown in Figure 27 and the input layer node name of the BP neural network is shown in Table 2.

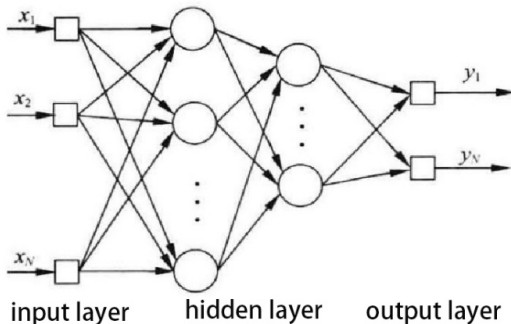

**Figure 27.** BP network structure diagram.

**Table 2.** Input layer node name.

| Input Layer Node Number | 1 | 2 | 3 | 4 | 5 | 6 |
|---|---|---|---|---|---|---|
| **Input layer node name** | signal duration | maximum amplitude | main frequency | the signal fractal dimension | IMF1 level energy to the total energy | IMF2-5 level energy to the total energy |

Because there are few analysis variables, the hidden layer of the BP neural network is selected. When the number of hidden layer nodes satisfies the following three empirical formulas at the same time, the number of hidden layer nodes is relatively accurate.

(1)

$$\sum_{i=0}^{n} C_M^i > k \tag{12}$$

*k⁻number of samples; M⁻hop count; n⁻number of input layer neurous; if i > M, so $C_M^i = 0$.*

(2)

$$M = \sqrt{n + m} + a \tag{13}$$

*n⁻number of input layer neurous; m⁻number of output layer neurons; a⁻the constant between $[0, 10]$.*

(3)

$$M = \log_2 n \tag{14}$$

*n⁻number of input layer neurous.*

According to the above formula, the hidden layer M of the BP neural network can be a constant between 9 and 13, the number of cycles is set to 1000 times, and the deviation of data from different nodes is selected. It can be analyzed that the deviation is the smallest when *M* = 9, so the number of hidden layer nodes is selected as 9 nodes.

By setting the rock fracture signal output as (1, 0) and the rock blasting signal output as (0, 1), the output result processing method determines the column where the maximum output data is, and the maximum column becomes 1 and the other column becomes 0.

The time–frequency characteristics (signal duration, maximum amplitude, main frequency), energy distribution characteristics, and fractal feature parameters of the collected 50 rock fracture signals and 50 rock blasting signals are used as the characteristic parameter database of rock fracture and blasting events. In each training, 30 rock fracture signals and 30 rock blasting signals are randomly selected as learning samples by MATLAB, and the remaining data are used as test samples. The weight adjustment method adopts the steepest descent method, and the BP network is used to learn the sample set. After the learning is completed, the remaining 20 rock fracture signals and rock blasting signals are identified. A total of 10 model trainings were performed. The results of each training are shown in Table 3.

**Table 3.** 10 training recognition accuracy statistics.

| Learning and Testing | 1 | 2 | 3 | 4 | 5 | 6 | 7 | 8 | 9 | 10 |
|---|---|---|---|---|---|---|---|---|---|---|
| Rock fracture signal recognition accuracy | 95.24% | 100% | 90.91% | 95.45% | 88.24% | 82.35% | 94.12% | 100% | 100% | 100% |
| Rock blasting signal recognition accuracy | 94.73% | 100% | 94.44% | 83.33% | 95.65% | 95.65% | 95.65% | 78.95% | 100% | 95% |

The BP neural network finally has a judgment accuracy of 94.63% for microseismic signals of rock fracture events and 93.34% for microseismic signals of rock blasting events.

## 6. Conclusions

This study introduces a method for extracting the eigenvalues of microseismic signals. By comparing the two types of signals, the differences in time–frequency, energy distribution, and fractal characteristics were analyzed. The following conclusions are drawn:

1.  The duration of rock fracture signal was mainly distributed in the range of 0–100 ms, and the maximum amplitude was mainly concentrated in the range of 0–500 mv. The main frequency was mainly distributed in the section above 500 Hz, the main energy band was IMF 1, and the fractal box dimension (*D*) below 1.1 accounted for 88% of the samples.
2.  The duration of rock blasting events was distributed in the range of above 50 ms, and the maximum amplitude of the signal had no definite range. The main frequency was mainly distributed in the range of 0–500 Hz; the main energy bands were IMF 2, IMF 3, and IMF 4, and the fractal box dimension (*D*) was more than 1.25, accounting for 70% of the samples.
3.  An automatic analysis and recognition system for microseismic signals based on the analysis of signal eigenvalues is established. The system structure is simple, and the judgment system is simple and clear, which can flexibly change the judgment basis of the system according to the actual situation of different mines and has high judgment accuracy. For the microseismic signal extracted from the mine selected in this paper, the recognition accuracy of rock fracture signal was 94.63%, and that of rock blasting signal was 93.34%. Compared with the single method, the recognition accuracy was improved.
4.  The system has the advantages of simple and easy acquisition of signal feature recognition. For different types of mines, it is easy to modify the recognition model according to the actual signal characteristics collected on site, so it has a wide range of applicability. Rock microseismic signals and rock fracture signals represent different change rates of rock mass. Therefore, accurate distinction between rock microseismic signal and rock fracture signal has a wide range of applications in the field of underground mine disaster prevention and control.

**Author Contributions:** Methodology, J.C., H.L., C.R. and F.H.; formal analysis, J.C., H.L., C.R. and F.H.; investigation, F.H.; resources, J.C. and C.R.; data curation, J.C., H.L., C.R. and F.H.; writing-original draft preparation, J.C., H.L., C.R. and F.H.; writing-review and editing, J.C., H.L., C.R. and F.H.; supervision, J.C. and C.R.; funding acquisition, J.C. and C.R. All authors have read and agreed to the published version of the manuscript.

**Funding:** This work was funded by the Natural Science Foundation of China (U1602232).

**Institutional Review Board Statement:** Not applicable.

**Informed Consent Statement:** Not applicable.

**Data Availability Statement:** Not applicable.

**Acknowledgments:** We would like to thank the editor and anonymous reviewers for their constructive comments. We would like to thank the staff of Dahongshan Iron Mine for their help in signal collection.

**Conflicts of Interest:** The authors declare no conflict of interest.

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
