# Peer review of "Automatic Identification System for Rock Microseismic Signals Based on Signal Eigenvalues"

_applsci, doi:10.3390/app13042619_

Round 1
Reviewer 1 Report
I have read the paper, which investigated what would distinguish rock blasting and rock fracture microseismic signals accurately, employing Hilbert Huang transform (HHT) as a signal analysis method, fractal box dimension, and BP neural network is commendable, however, these are my comments;
11. The objectives and the rationale of the study clearly stated are not sufficiently stated. The authors should rewrite the abstract, to state briefly the purpose/objective/problem statement of the research, the principal results, and major conclusions. What is the benefit of distinguishing rock blasting and rock fracture microseismic signals?
2. The introduction should be completely rewritten to bring out the research gap, citing more references, highlighting the necessity of the research, and mentioning how your study is going to address any/some of the gaps/novelty.
3. A subheading should be written explaining the methods used with all the necessary steps.
4. Aside from the aim stated in the title, the research gap and the goals of the research are not specified which leads to the reader missing the significant contribution of the research.
5. The conclusion part should enhance your contributions, and limitations, underscore the scientific value added of your paper, and/or the applicability of your findings/results and future study in this session or the conclusion should be enhanced by describing how this study is unique and distinct from others, as well as outlining the primary findings. Furthermore, other recommendations should be suggested for future studies if applicable.

Author Response
Dear Expert :
1、For your first expert opinion
In the abstract part, I explain the reasons for the research and its benefits and explain the main conclusions.
2、For your second expert opinion and fourth expert opinion
I rewrote section 1.1 of the introduction, citing more references, highlighting the importance of research and research innovation. At the same time, the purpose of the study and the shortcomings of the existing research are stated in this section.
3、For your third expert opinion
I added a 1.2 section in the first section to illustrate the methodology used in this study and all the necessary steps.
4、For your fifth expert opinion
I added a fourth conclusion in the conclusion section of section 6 to illustrate the scientific value of my paper and the applicability of the study to this meeting.
Reviewer 2 Report
Very interesting work about how to automatically identify if a signal comes from rock blasting or fracture.
I believe it would benefit on explaining with more detail each one of the methods used to identify signals.
Some figures should be easier to read
Check table 1 (data missing)
Does band Energy represent any physical property? Please explain.
Additional information about method used to estimate fractal dimension is necessary.
Please explained how the training data is chosen. What do you mean with group of data? Is it just one signal or more?

Author Response
Dear Expert :
- For your first expert opinion
I distinguish the part of the chart that explains the chart symbols from the statistical symbols by boxes.
- For your second expert opinion
I checked the data in Table 1 and supplemented the missing part of the data.
- For your third expert opinion
In the fifth paragraph of the third chapter, I add the explanation of the physical meaning of IMF.
Specific as follows:
EEMD is used to decompose the de-noised signal into 8 layers. After decomposition, a total of 9 different spectrograms of IMF1-IMF8 and remainder are generated, which are arranged from high to low frequency. Corresponding to the IMF1-IMF8 and remainder, a total of 9 corresponding frequency band energy are generated. The ratio of energy of each frequency band to the total frequency band energy represents the proportion of different spectrograms in the de-noised signal after decomposition.
- For your fourth expert opinion
In the fourth section, I add 4.1 to explain how to calculate the fractal dimension in detail.
- For your fifth expert opinion
I added a seventh paragraph in Section 5 to detail how I trained the recognition model.